# Fabrication of Double Emission Enhancement Fluorescent Nanoparticles with Combined PET and AIEE Effects

**DOI:** 10.3390/molecules25235732

**Published:** 2020-12-04

**Authors:** Hsing-Ju Wu, Cheng-Chung Chang

**Affiliations:** 1Research Assistant Center, Show Chwan Memorial Hospital, Changhua 500, Taiwan; hildawu09@gmail.com; 2Department of Biology, National Changhua University of Education, Changhua 500, Taiwan; 3Graduate Institute of Biomedical Engineering, National Chung Hsing University, No.145, Xing Da Road, Taichung 402, Taiwan; 4Intelligent Minimally-Invasive Device Center, National Chung Hsing University, No.145, Xing Da Road, Taichung 402, Taiwan

**Keywords:** fluorescent silica nanoparticles (FSN), double emission enhancement (DEE), Stöber method, photoinduced intramolecular electron transfer (PET), aggregation-induced emission enhancement (AIEE)

## Abstract

The major challenge in the fabrication of fluorescent silica nanoparticles (FSNs) based on dye-doped silica nanoparticles (DDSNs) is aggregation-caused fluorescence quenching. Here, we constructed an FSN based on a double emission enhancement (DEE) platform. A thio-reactive fluorescence turn-on molecule, *N*-butyl-4-(4-maleimidostyryl)-1,8-naphthalimide (CS), was bound to a silane coupling agent, (3-mercaptopropyl)-trimethoxysilane (MPTMS), and the product *N*-butyl-4-(3-(trimethoxysilyl-propylthio)styryl)-1,8-naphthalimide (CSP) was further used to fabricate a core–shell nanoparticle through the Stöber method. We concluded that the turn-on emission by CSP originated from the photoinduced electron transfer (PET) between the maleimide moiety and the CSP core scaffold, and the second emission enhancement was attributed to the aggregation-induced emission enhancement (AIEE) in CSP when encapsulated inside a core–shell nanoparticle. Thus, FSNs could be obtained through DEE based on a combination of PET and AIEE effects. Systematic investigations verified that the resulting FSNs showed the traditional solvent-independent and photostable optical properties. The results implied that the novel FSNs are suitable as biomarkers in living cells and function as fluorescent visualizing agents for intracellular imaging and drug carriers.

## 1. Introduction

Silica nanoparticles (SNs) have many functions that make them superior to other similar polymers, such as ease of synthesis and surface modification, chemical and physical stability, biocompatibility and excellent dispersibility in aqueous media [1]. In particular, SN is nonluminescent and chemically inert, so it is to expect a low environment impact and is therefore an ideal material for coating nanomaterials, encapsulating molecules or drug carriers [2]. Thus, research on dye-doped silica nanoparticles (DDSNs) has been rapidly gaining attention in the past two decades [3]. Researchers have a great interest in constructing DDSNs with controllable size, high emission intensity and good photostability at physiological pH for potential applications in many nanotechnology fields [4]. When DDSN-embedded dyes are replaced by a fluorophore, fluorophore-doped silica nanoparticles (FSNs) are obtained, which offer all the diagnostic and theragnostic features required for their nanotechnological application in biological fields [5,6]. Compared with luminescent dye-doped latex nanoparticles [7,8], intrinsically luminescent nanomaterials [9] or quantum dots (QDs) [10], FSNs are very successful tools as fluorescent emitters in nanotechnological applications. Here, we discuss FSNs as fluorophore-doped inner silica nanoparticles of DDSN without any modification of the silica nanoparticle surface with a fluorescent dye.

Since great progress has been made in preparing DDSNs, FSNs can be prepared by incorporating fluorophores into the silica networks or their shells via physical processes or chemical reactions [3]. At present, versatile silane technology can not only encapsulate fluorophores with excellent optical properties but also accurately adjust the shape and size to produce an ideal nanomaterial. Considering their successful application in biological research [10,11], FSNs are applied in fluorescent sensors [12,13], as biomarkers [14,15] and even for diagnosis [16]. In particular, in vivo imaging is the leading biological application of FSNs [17,18]. The encapsulation of designated fluorophores determines the optical properties of the FSNs and their application, such as multicolor imaging [19] or NIR imaging [20,21]. As we expected, when fluorophores are incorporated into silica nanoparticles, FSN possesses a better photostability because the silica matrix (or shell) protects the fluorophores from oxygen and other harmful species, which may cause photobleaching. However, in most cases, it is difficult to obtain a highly luminescent FSN via common encapsulation methods due to their aggregation-caused quenching (ACQ) effect. The ACQ and their poor photostability in aqueous media are two common problems of organic fluorescence dyes that cause a dramatic reduction in the fluorescence imaging quality. Successful FSNs must overcome this critical problem of ACQ caused by a high density of fluorophores inside the FSN core. Thus, some effective strategies based on molecular construction have been proposed. One strategy introduces aggregation-induced emission enhancement (AIEE) molecules, especially functionalized AIEE molecules, to form FSNs, which are then coated with a silane coupling agent [21,22,23]. AIEE is a process that contrasts with the ACQ effect, in which nonemissive luminogens are induced to emit light by aggregate formation based on a working mechanism of restricted intramolecular rotation (RIR) [24]. Another strategy employs a local hydrophobic cage inside the FSN, in which a long-chain hydrophobic silane coupling agent is introduced into the FSN to prevent the π−π stacking of fluorophores and inhibit the twisted intramolecular charge transfer (TICT) or RIR to prevent the ACQ effect and inward diffusion of water molecules [25,26,27].

In a previous study, we developed a 1,8-naphthalimide-based fluorescence turn-on probe CS containing a maleimide group for the detection of biological thiols. This probe can not only be used to detect intracellular thiol in living HepG2 cells but also dengue virus via fluorescent imaging [28]. The purpose of this research is to combine the photoinduced electron transfer (PET) principles and the characteristics of AIEE to develop a nanosystem with double fluorescence enhancement. First, biosensor CS was bound with thiol-modified supported silane coupling reagent MPTMS (mercapto-propyltrimethoxysilane) to achieve a fluorescence enhancement by PET. Then, the Stöber method was used to prepare FSNs with a fluorescent core and a silica shell, which promoted the second emission enhancement based on AIEE. The study confirmed that this FSN was optically stable, and its fluorescent performance was not affected by aqueous media or most organic solvents. Further surface modification can effectively improve the water solubility and the successful uptake by living cells. The results imply that these new fluorescent nanoparticles are suitable as new fluorescent materials and can be used in intracellular imaging and as biomarkers.

## 2. Results and Discussion

### 2.1. Material Design and DEE

Maleimides are known to readily undergo a selective electrophilic Michael addition reaction with thiol groups and are responsible for emission switching in the fluorescence system [29]. When RS-H undergoes addition across the C=C bond of the maleimide unit to disrupt the n–π electronic conjugation, it thereby hinders the PET and then causes fluorescence enhancement. Thus, maleimide- coupled fluorophores are known to be fluorescence turn-on probes, whose fluorescence increases dramatically upon reaction with biothiols [30,31,32]. Based on this behavior, we prepared a cysteine probe by linking a maleimide to naphthalimide fluorophores. The starting material *N*-butyl-4-bromo-1,8-naphthalimide was coupled with vinyl aniline by means of the Heck reaction to generate the intermediate *N*-butyl-4-(4-aminostyryl)-1.8-naphthalimide and then converted to the target compound CS with maleimide. Since CS successfully and rapidly reacts with cysteine and is used to label appropriate amino acids in dengue virus proteins and light-on single dengue virus spots in cells [28], we carried out the thiolate reaction of CS with MPTMS to generate CSP in a high yield above 90%. CSP, a precursor of FSNs, was expected to retain the emission turn-on optical property due to the decrease in PET, which is the first emission enhancement of DEE. Furthermore, we also found that CSP can form fluorescent organic nanoparticles (FONs) based on AIEE, which means that CSP should also induce AIEE when covalently grafted to the nanoparticle silica network and encapsulated in the core of FSNs. Thus, the DEE distinguishing feature based on the first emission enhancement caused by PET disappeared, and the second emission enhancement was attributed to the AIEE during core–shell silica nanoparticle formation (Figure 1).

### 2.2. PET and AIEE

The absorption and fluorescence spectral properties of CS in the absence and addition of thiol reagents are shown in Figure 2a. Upon the addition of mercaptan (ethanethiol, cysteine, MPTMS, 10 eq.) to the CS-containing ethanol solution, the absorbance peak at 390 nm and the patterns did not change much, while the emission intensities increased with quantum yields from ~0.021 (510 nm) to ~0.5, which was caused by decreasing PET from the core scaffold to the maleimide moiety in the excited state. This high signal-to-noise ratio (~35-fold) of the thiolation products allowed us to prepare a fluorophore by employing a silane coupling reagent to construct a bright DDSN emitter. Thus, we prepared compound CSP, and its solvent effect on the fluorescence response was evaluated as shown in Figure 2b. It is worth noting that the emission enhancement from CS to CSP occurred both in organic solvents and aqueous solvents.

Unexpectedly, a higher quantum yield of CSP was observed in polar (protic) solvents, except for water. In previous studies, we investigated the AIEE properties to discuss the fluorescence behavior of solid-state and aqueous fluorogens, and we observed the formation of fluorescent organic nanoparticles (FONs) [33,34,35]. Briefly, the AIE phenomenon was easily detected in the emission spectra in THF/H_2_O mixed solvents, as shown in Figure 3, which was also supported by the fluorescence microscopy and SEM images (Figure 3d,e). The SEM image of the Fs prepared from a related mixed solvent clearly shows nanoparticles with a mean diameter ranging 50~120 nm, which were CSP concentration-dependent. Here, we prove that CSP can self-assemble to form nanoparticles and that the thermal relaxation of the excited state was suppressed due to the restriction of intramolecular rotation when CSP aggregated inside the nanoparticles. Under this condition, the flat molecules can intermolecularly stack with each other in the concentrated or solid state, resulting in an emissive non-resonance excitonic state of H-aggregates or emissive J-aggregates. Usually, these types of dyes are emissive in organic solvents as well as in the solid state, even though they exhibit a π-stacking pattern. Therefore, CSP should not be regarded as an AIEgen but rather as a “nonresonant exciton system” facilitating the AIEE and causing the CSP-containing nanoparticles to become emitter FONs [36,37,38]. Nevertheless, this result provides evidence that CSP possesses the emissive-aggregate potential to form FONs, which is the reason for the brighter fluorescence of powdered CSP. Furthermore, the FONs are suitable as a core for silane coupling agents used to coat the shell by the Stöber method. Thus, we expect that the bare FONs can be reinforced for use as fluorescent dots by silane coating to form FON-doped silica nanoparticles.

### 2.3. Construction of FSNs

As we ensured that the CSP could form FONs, ACQ should not occur upon CSP aggregation inside the nanoparticles. We propose that the AIEE-characteristic CSP, containing a silane coupling moiety, should be suitable for generating FSN based on the DDSN principle. To construct the CSP-based core–shell-structured nanoparticle FSN, another silane coupling reagent, tetraethyl orthosilicate (TEOS), was used to co-hydrolyze with CSP and coat the surface of the resulting FSN through a modified Stöber method, affording CSP-SN@OH with a hydroxyl-terminated surface (Figure 1, pathway iii). As shown in Figure 4a, in the preparation procedure, further emission enhancement was observed when the silane coupling reaction was performed by adding TEOS and forming CSP-SN@OH. The CSP-SN@OH nanoparticle product was characterized by SEM and fluorescence microscopy, as shown in Figure 4b,c, respectively. Meanwhile, the size distribution located at 80 nm (Min ~57 nm, Max ~ 105 nm), by collecting it from the dynamic light scattering (DLS) (Appendix A), which revealed a broad dispersion in size, thus we extracted the size of NWs from the statistical analysis of several SEM images to identify 60–70 nm of product CSP-SN@OH. Since the CSPs were encapsulated in the core of a silanol surface nanoparticle in an aggregated state, the obtained FSN@OH was expected to emit strong fluorescence due to the AIEE, as mentioned above. Therefore, compared with the solvent effect of the CSP monomer (Figure 2b), Figure 4d shows the persistent emission without any solvent effect when CSP-SN@OH was dispersed in water and other solvents. This result indicated that the CSP fluorophore was perfectly encapsulated and protected inside the poly siloxane shell, resulting in unique optical properties, which satisfactorily met the requirements of superior biomarkers and biotags. To optimize the construction conditions of CSP-SN@OH, we observed that the emission enhancements and sizes of the produced nanoparticles became more spherical and uniform as the amount of TEOS added increased from 100 to 1000 μL (Appendix A), which was accompanied by a redshift of the spectra from 510 to 540 nm, with respect to the CSP in ethanol.

Furthermore, the Lippert plots of CS, CSP and CSP-SN@OH were obtained and provided further insight (Appendix A). It is known that the Lippert slope represents the intramolecular charge transfer (ICT) or TICT variation degree of the molecule to the solvent as well as the sensitivity of the molecule to the environment. We observed that the Lippert slope (stock shift ∆ν to solvent polarity ∆*f*) of CSP-SN@OH was flatter than that of CS and CSP, which indicates that CS and CSP showed similar and apparent ICT changes, while the produced nanoparticles were not affected by the solvent. The result is consistent with Figure 2b and Figure 4d. Moreover, the redshifted emission of CSP-SN@OH could be attributed to the restricted intramolecular rotation mechanism-caused rigid-TICT state molecular conformation of CSP in nanoaggregates [35]. For example, the coplanar-to-twist molecule geometry restricted the inner core of the nanoparticle. In summary, the optical property change from CS to CSP was attributed to PET rather than ICT, and the change from CSP to CSP-SN@OH was attributed to ICT (TICT) caused by restricted intramolecular rotation to achieve the AIEE.

### 2.4. Surface Modification and Cellular Imaging

We incubated A549 lung cancer cells with the CSP-SN@OH nanoparticles, which were not easily taken up by the cells. Surface modification was performed with a similar procedure following the construction of CSP-SN@OH but using a mixture of APTMS/TEOS (Figure 1 pathway iv). We also evaluated the effect of the mixing ratio of these two silane-coupling agents (TEOS/APTMS = 8/2, 7/3, 6/4 and 5/5 in 1000 μL total volume) on the nanoparticle construction (Appendix A). It was found that the product did not easily form spherical nanoparticles until the ratio of TEOS/APTMS exceeded 7/3; increasing the ratio of APTMS resulted in larger products and more complex products. Finally, inter-sphere silane coupling reactions occurred when the TEOS/APTMS ratio was greater than 5/5. The characterization of the sizes, emission wavelengths and solvent-independent effects of these APTMS-modified nanoparticles is shown in Appendix A and Figure 5. We chose the nanoparticle constructed at a ratio of 7/3 as our product nanoparticle CSP-SN@OH/NH_2_, whose spectral properties are consistent with those of CSP-SN@OH, but better water solubility and dispersion in the solvent were observed. It is worth noting that this nanoparticle was affected by the solvent to some extent, which was attributed to possible interlaced cross-coupling on the TEOS/APTMS hybrid surface to achieve a larger gap, through which solvent molecules can penetrate the shell of the nanoparticle.

We evaluated the cellular uptake and cell viability of the nanoparticle and found that the cellular uptake of the CSP-SN@OH/NH_2_ nanoparticles was higher than that of the CSP-SN@OH nanoparticles. Figure 5 clearly reveals that CSP-SN@OH/NH_2_ stayed in the cytoplasm of the A549 cells. This result indicates that the amine-modified silica nanoparticles presented higher cellular uptakes compared with the uptake of the hydroxylated silica nanoparticles. Therefore, we also evaluated the cellular cytotoxicity of CSP-SN@OH/NH_2_ in human cancer cell lines HeLa (cervical) and A549 (lung) and human normal cell line MRC-5 (lung), and low cytotoxicity (over 90% normal cell viability, 5 mg) was observed for 24 h (Appendix A). Eventually, continuing the experiment in Figure 5e, we benchmarked the photostability of the produced nanoparticles against those cell markers for long-term live-cell imaging. To perform this comparison, CSP-SN@OH/NH_2_, MitoTracker™ Red (mitochondria tracker), lysotracker green (lysosome tracker) and Nile Red (lipid droplet tracker) incubated in A549 cells were continuously imaged for over 300 s, and then the emission intensity was monitored under the same light source and region of interest (ROI). In Appendix A, CSP-SN@OH/NH_2_ cell staining images present the most photostability over the other trackers, and surprisingly, the photostability of fluorescence remained unchanged for more than 250 s. These superiorities of the produced nanoparticle make us look forward to more functional applications where these FSNs (CSP-SN@OH or CSP-SN@OH/NH_2_) will be coupled with organic drugs or biosensors, making them a very powerful tracing tool for drug carriers.

## 3. Experiment 

### 3.1. Materials

The chemicals tetraethyl orthosilicate (TEOS), Mercaptopropyl)trimethoxysilane (MPTMS) and ammonium hydroxide solution (25%) employed in this study were of the best analytical grade available and obtained from Sigma-Aldrich Chemical Co. (St. Louis, MO, USA) or Merck Ltd. (Darmstadt, Germany), and were used without further purification. All the solvents for spectral measurements were of spectrometric grade and other reagents were purchased from Aldrich and used as received.

### 3.2. Apparatus

The absorption and fluorescence spectra were generated using a Thermo Genesys 6 UV-visible spectrophotometer (Thermo Fisher Scientific Inc, Waltham, MA, USA ) mounted on a double beam sample detecting with a 1 nm resolution and a HORIBA JOBIN-YVON FluoroMax-4 spectrofluorometer (HORIBA Ltd., Kyoto, Japan), with a 1 nm bandpass filter in a 1 cm cell length at room temperature, respectively. The fluorescence images were taken under a Leica AF6000 fluorescence microscope with a DFC310 FX digital color camera (Major Instruments Co., Ltd., New Taipei City, Taiwan). The SEM images of the nanostructures were obtained using a JEOL JSM-7800F Prime Schottky field emission scanning electron microscope (JEOL Ltd. Tokyo, Japan). Dynamic light scattering (DLS) measurement was recorded using a SZ-100–HORIBA (HORIBA Ltd., Kyoto, Japan).

### 3.3. Cell Culture Conditions and Cellular Imaging

MRC-5 (human normal lung fibroblast cell line, BCRC, Hsinchu, Taiwan) and HeLa (human cervical carcinoma cells, BCRC, Hsinchu, Taiwan) used CORNING^®^ MEM (Minimum Essential Medium) as the culture medium, and the A549 human lung adenocarcinoma cancer cell (BCRC, Hsinchu, Taiwan) line was grown in RPMI 1640 (Gibco/Invitrogen, Cat. No. 22400-089) medium containing 10% fetal bovine serum (FBS). The cells were cultured at 37 °C in a humidified atmosphere (95% air and 5% CO_2_) and then seeded onto coverslips and incubated for 24 h before observation by cellular imaging. On the next day, cells were incubated with nanoparticles for 12 h; here, the stock solutions of the compounds were diluted in serum-free medium before use (1:100 *v*/*v*). The fluorescence images were taken under Leica AF6000 fluorescence microscopy with a DFC310 FX digital color camera. The excitation sources were 390/10 or 470/20 nm bandpass for compounds. Fluorescence photographs were taken through related ranges tubes.

### 3.4. Synthesis of N-Butyl-4-(3-(Trimethoxysilyl-Propylthio)Styryl)-1,8-Naphthalimide (CSP)

To prepare CSP as an FSN precursor, as depicted in Figure 1, previous thiol sensor *N*-butyl-4-(4-maleimidostyryl)-1,8-naphthalimide (CS) (Appendix A) [28], was used. In a typical procedure, in an oven-dried 25 mL round-bottom flask, CS was taken (0.2 g, 0.5 mmol) in 10 mL EtOH. Then, 3-Mercaptopropyl)trimethoxysilane (MPTMS, Sigma-Aldrich, 1.96 g, 10 mmol) was added to the system, and the resultant mixture was stirred for 2 h at room temperature. The reaction mixture was concentrated under high vacuum to a trace amount and hexane was poured to recrystallize the product yellow powder CSP. Yield 0.26 g, 81%. ^1^H-NMR (400 Hz, DMSO-*d*6): δ 9.01 (d, *J* = 8.0 Hz, 1H), 8.54 (d, *J* = 8.2 Hz, 1H), 8.50 (d, *J* = 8.0 Hz, 1H), 8.29 (d, *J* = 8.2 Hz, 1H), 8.28 (d, *J* = 16.0 Hz, 1H), 7.99 (d, *J* = 7.8 Hz, 2H), 7.93 (dd, *J* = 8.0, 7.8 Hz, 1H), 7.65 (d, *J* = 16.0 Hz, 1H), 7.35 (d, *J* = 7.8 Hz, 2H), 4.10 (t, 2H), 4.04 (t, 2H), 3.45 (s, 9H), 2.47 (m, 2H), 1.62 (m, 2H), 1.39 (m, 2H) 1.35 (m, 2H), 0.91 (t, 3H), 0.71 (m, 2H) ppm. HRMS (ESI, *m*/*z*): [M + H]^+^ 644.81; found, 645.2. Anal. Calcd for CSP C_34_H_36_N_2_O_7_SSi: C, 63.33, H, 5.63; N, 4.34; CSP·0.5H_2_O: C_34_H_37_N_2_O_7.5_SSi; found: C, 62.46; H, 5.70; N, 4.21.

### 3.5. Determination of Quantum Yields

The quantum yields of PTZ derivatives were determined according to the literature [39].
Φu = Φs × (Afu × As × λexs × ηu)/(Afs × Au × λexu × ηs)
where Φu is the quantum yield of unknown; Af is the integrated area under the corrected emission spectra; A is the absorbance area at the excitation wavelength; λex is the excitation wavelength; η is the refractive index of the solution; and the subscripts u and s refer to the unknown and the standard, respectively. Here, we used Rhodamine Green as a standard and for the same λex, we chose BMVC as the standard, which has the quantum yield of 0.25 in glycerol and 0.02 in DMSO [40].

### 3.6. Construction of CSP-SN@OH

In brief, 7.5 μmol of CSP in 0.1 mL of DMSO was slowly added to a 15 mL ethanol solution under ultrasound irradiation. Then, 0.195 mL of distilled water and 0.32 mL of ammonium hydroxide (25%) were added into the mixture. The solution was stirred at room temperature for approximately 30 min, after which 100, 500 or 1000 μL of TEOS in 3.5 mL of an ethanol solution were added dropwise into the mixture and stirred at room temperature for another 24 h. Finally, the mixture was centrifuged (6000 rpm/20 min) and washed with water and ethanol three times to remove the unencapsulated compound.

### 3.7. Construction of CSP-SN@OH/NH_2_

Similar to the process for CSP-SN@OH, 7.5 μmol of CSP in 0.1 mL of DMSO was slowly added to a 15 mL ethanol solution under ultrasound irradiation. Then, 0.195 mL of distilled water and 0.32 mL of ammonium hydroxide (25%) were added into the mixture. The solution was stirred at room temperature for approximately 30 min, after which (0.8/0.2, 0.7/0.3, 0.6/0.4 or 0.5/0.5 mL/mL) of TEOS/APTMS 1 mL in 3.5 mL of an ethanol solution were added dropwise into the mixture and stirred at room temperature for another 24 h. Finally, the mixture was centrifuged (6000 rpm/20 min) and washed with water and ethanol three times to remove the unencapsulated compound.

## 4. Conclusions

In this manuscript, the PET inhibition-induced fluorescence enhancement phenomenon was promoted according to the AIEE effect, which can be used to build a novel DEE platform. The room temperature-reacted thiol biosensor CS was silanized with MPTMS to produce CSP, which was the first stage of the emission enhancement due to the inhibition of PET. After that, another silane coupling agent was further added to coat a silica shell around the CSP core with the Stöber method. The emission of the obtained silica nanoparticles CSP-SN@OH was expected to increase further with AIEE. Systematic investigations verified that the resulting CSP-SN@OH showed the traditional solvent-independent and photostable optical properties in aqueous media as well as in most organic solvents. This result implies that these novel fluorescent nanoparticles are suitable as biomarkers in living cells and function as fluorescent visualization agents for intracellular imaging and drug carriers. There are several issues to address in our study. 1: The suppression of the molecular TICT process can induce emission enhancements, which may arise from a mechanism of self-assembly (present AIEE) and/or the “non-resonant exciton system” of the molecule. 2: The compound CSP simultaneously possessed AIE and PET properties and could form FONs, which are suitable for the FSN fabrication of DDSNs. 3. This superior property allowed us to construct a double emission enhancement silica nanoparticle.

## Figures and Tables

**Figure 1 molecules-25-05732-f001:**
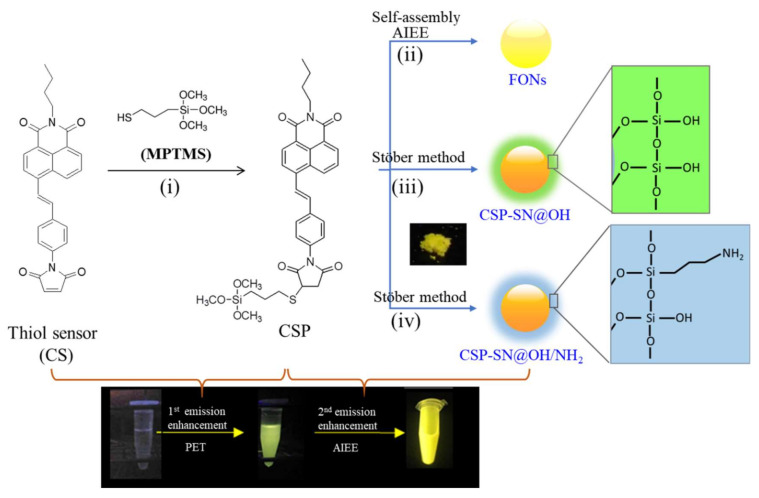
Synthesis of CSP and illustration of fluorescent organic nanoparticles (FONs), CSP-doped silica nanoparticles with a TEOS hydrolysis shell (CSP-NP@OH) and CSP-doped silica nanoparticles with a TEOS/APTMS mixed hydrolysis shell (CSP-NP@OH/NH_2_). (i) MPTMS, rt, 1 h; (ii) THF/H_2_O mixed solvent; (iii) TEOS, C_2_H_5_OH, NH_4_OH; (iv) TEOS, APTMS, C_2_H_5_OH, NH_4_OH.

**Figure 2 molecules-25-05732-f002:**
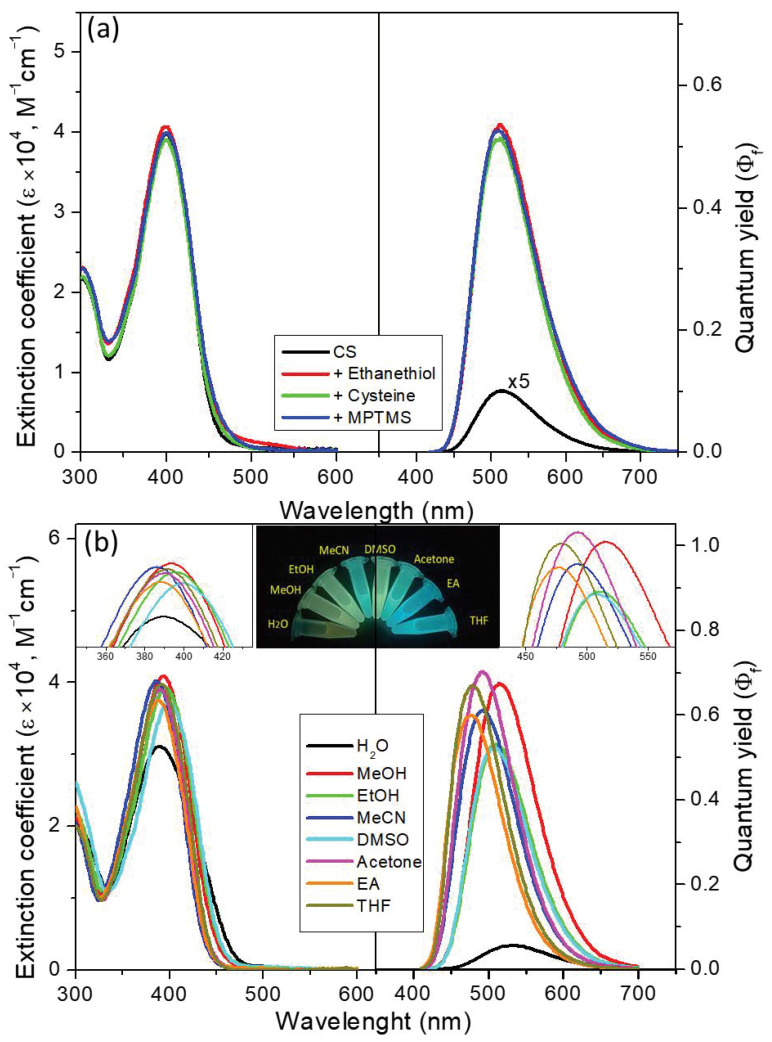
Absorbance (left) and emission (right) spectra represented by extinction coefficient and quantum yield, respectively. (**a**) Thiol-induced CS in ethanol; (**b**) solvent effect of compound CSP excited at the wavelengths of the absorption maxima. The insert spectra show the zoom in of tops of absorption/emission spectra, and the inset photos show the emission color changes of the probes (10 μM) under a handheld ultraviolet lamp (365 nm).

**Figure 3 molecules-25-05732-f003:**
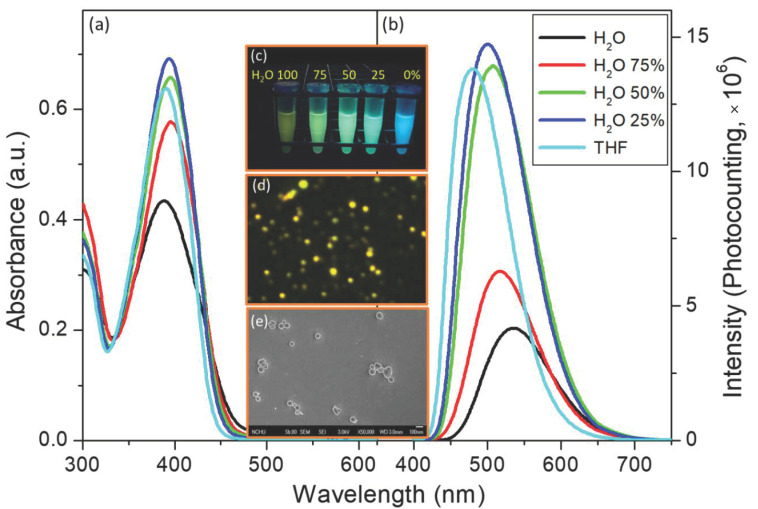
(**a**) Absorbance and (**b**) emission spectra of CSP in THF/H_2_O mixed solvents of various ratios. The inset photos show (**c**) the emission color changes under a handheld ultraviolet lamp (365 nm); (**d**) the fluorescence microscopy image (a 390 ± 20 nm bandpass filter as the excitation light source, with emission collected through a 410 nm long-pass filter); and (**e**) the SEM images of CSP FONs prepared from THF/H_2_O = 1/1, *v*/*v* (25 μM).

**Figure 4 molecules-25-05732-f004:**
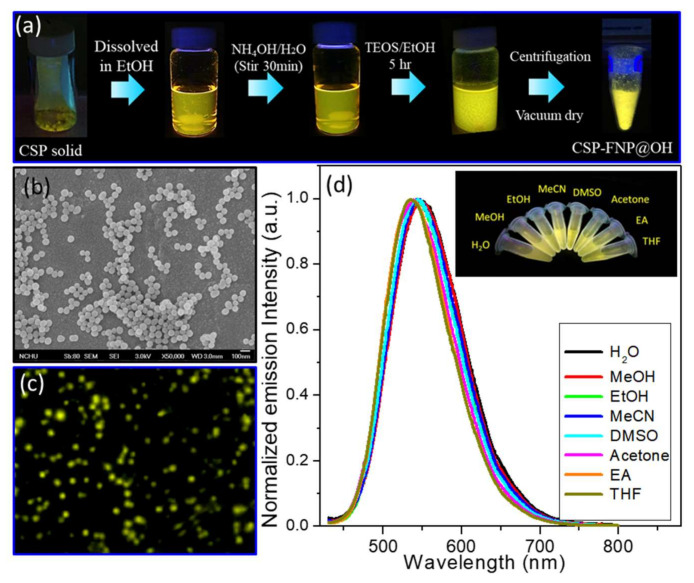
(**a**) Preparation scheme of CSP-SN@OH from CSP; (**b**) SEM images (scale bar 100 nm); (**c**) fluorescence microscopy image (Ex: 390 ± 20 nm bp filter; Em: 410 nm lp filter) and (**d**) solvent-independent emission spectra of CSP-SN@OH in various solvents. The inset photos show the emission color under a handheld ultraviolet lamp (365 nm).

**Figure 5 molecules-25-05732-f005:**
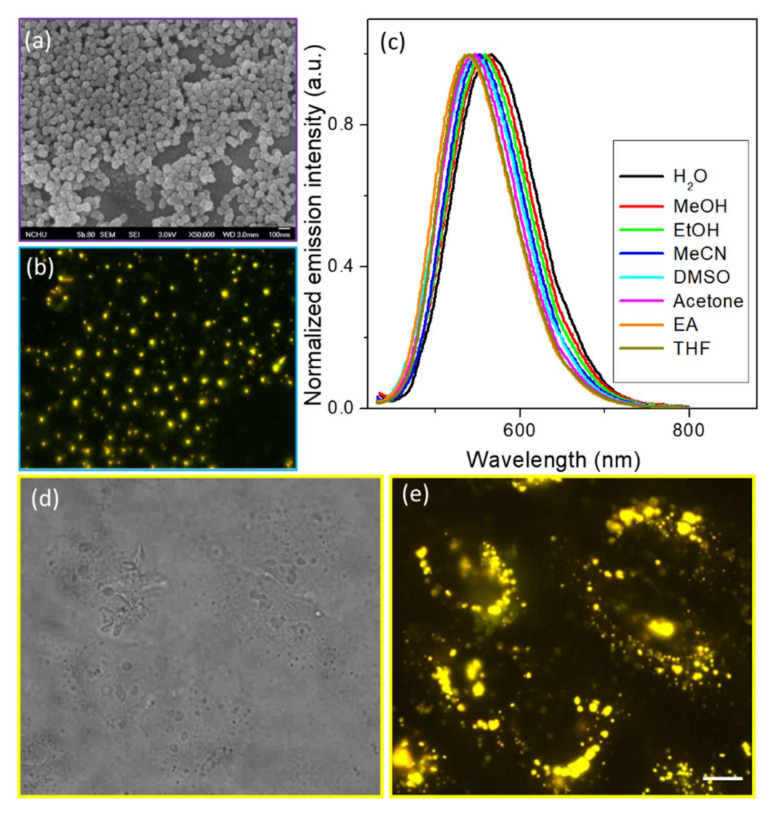
(**a**) SEM images (scale bar 100 nm); (**b**) fluorescence microscopy image (Ex: 390 ± 20 nm bp filter; Em: 410 nm lp filter); and (**c**) solvent effect emission spectra of CSP-SN@OH/NH_2_. Cellular imaging of CSP-SN@OH/NH_2_ (1 mg, incubated 12 h) in A549 cancer cells. (**d**) Brightfield. (**e**) UV cube (ex 390/10 nm, 410 long-pass filter). Scale bar = 10 μM.

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
