# Peer review of "Fabrication of Double Emission Enhancement Fluorescent Nanoparticles with Combined PET and AIEE Effects"

_molecules, 2020, doi:10.3390/molecules25235732_

Round 1

Reviewer 1 Report

The manuscript describes the development of fluorescent silica nanoparticles for biological applications. The study is well-designed and clearly presented, and should be suitable for publication after the following minor points have been addressed:

1) Page 1, line 34: 'harmless to the environment' this is a very strong statement that is very likely to turn out false in the future. It should be toned down, e.g. to 'expected low environmental impact'

2) line 55: 'that to increase the stability.' should be deleted

3) Coupling constants should be given to the nearest 0.5 Hz, so e.g. not 8 Hz, but 7.5 Hz or 8.0 Hz (as appropriate.)

4) line 95: 'Fluoromas' should be 'Fluoromax'

5) line 148: 'When a thiol group is present at the C=C double bond' should be replaced with something like: 'when RS-H addition across the C=C bond disrupts the n-pi electronic conjugation...'

6) 'conjugated' is used too indiscriminately, and should not be used for the joining of small molecule fragments in e.g. a Heck reaction. It should be reserved for the reactions of large (bio) molecules.

7) lie 167: 'basic' is confusing, please replace with 'absorption and emission in the absence of thiol'

8) line 182: 'protic' does not seem to be true, the quantum yield is lowest in water!

9) In general, the manuscript is too heavy on abbreviations, which makes it quite difficult to read in parts. I'd suggest that any abbreviation that is not used at least 3-4 times is simply written out in full. Examples are DV, RIR...

10) line 195: 'pi' should be replaced with the Greek letter.

11) line 270: there is no support for the NH2-modified particles being 'more biocompatible', as there are not toxicity data. Cellular uptake is neither a requirement not a hallmark of biocompatibility.

12) It would be helpful to have insets showing the tops of the absorption/emission spectra to clarify the order of maxima in the figures. At the moment it is not possible to distinctly see the spectra.

13) ESI: please provide the weights of the reagents and the products not just moles and % yields.

14) The C results in the CHN analyses are off in all places by a lot. Please comment.

Author Response

Dear Assistant Editor:

Dr, Aaron Yan

Enclosed please find a copy of the revised manuscript entitled “Fabrication of double emission enhancement fluorescent nanoparticles with combining PET and AIEE effects” for publication in Molecules.

We deeply appreciate you and referee’s kindness and patiently recommend us about the manuscript. The editor and reviewers’ comments are addressed in full as possibly as we can. First, we added several experimental results (Figure S2 Figure S6 and Figure S7) and redrawn Figure 2 and discussed some issues follow the suggestions from reviewers. Second, we corrected the mistakes, fulfilled some references for this time revision. All defences and revisions are marked with blue colour in manuscript and illustrated as follows.

I hope that you find this manuscript acceptable for publication in the article of Molecules. Please inform me if you have any other query and request.

Best regards,

Cheng-Chung Chang  20201202

8864-2284-0734#24,

ccchang555@dragon.nchu.edu.tw

Reviewer 2 Report

In the submitted paper the authors present a study on the fabrication of fluorescent silica nanoparticles which show enhanced emission. Here the authors demonstrated a procedure to produced dye-doped silica particles which did not suffer from aggregation-caused fluorescence quenching, instead their particles exhibit a “double” emission enhancement. This enhancement is assumed to be caused by photo-induced electron transfer (PET) and by a suppression of thermal relaxation of excited states in CSP when encapsulated inside a core-shell nanoparticle. As a results the authors claimed that they produced fluorescent nanoparticles exhibit solvent-independent and photo-stable optical properties which are well suited as biomarkers in living cells or as fluorescent visualizing agents for intra cellular imaging. The presented work describes an interesting and promising approach to obtain fluorescent silica nanoparticles with improved photo-physical properties. However, there are some points the authors need to consider before publication is possible:  

  • For non-expert readers the use of the high number of abbreviations used in the manuscript is inconvenient. Therefore, the respective abbreviations need to be introduced correctly. This did not happen for the following ones and need to be performed consequently at that position in the text where they are mentioned for the first time: CS, CSP, ICT, TICT, RIP, FON
  • The authors should present more evidence in their manuscript in which respect their fluorescent particles are beneficial/superior/complementary compared to other types of particles (QD, intrinsic luminescent nanoparticles) which are already used in the field.
  • Page 9, line 272-273: The authors should show/discuss in more detail how their particles where conjugated with drugs or biosensors to make them a powerful tool in life science applications.
  • In Fig. 2a the unit for the parameter on the y-axis and in Fig. S3 on the x-axis are missing.
  • For Fig. 2a,b the authors need to explain how the Quantum yield values were calculated (right y-axis) from the measured data. This calculation should be given in the “Experiment” section, together with more detailed information for the spectroscopic measurements: e.g. light-path lengths for absorption and emission measurements, give real OD values for absorption and state whether “inner Filter corrections” were necessary or not.
  • It is not stated explicitly in the manuscript which factor of enhancement in finally reached by the use of the DEE in this work. The authors should also provide explicit data/information on the long-time fluorescence emission stability. What is the smallest possible size of the silica particles and can one used them as tags for single-molecule studies?

Author Response

Dear Assistant Editor:

Dr, Aaron Yan

Enclosed please find a copy of the revised manuscript entitled “Fabrication of double emission enhancement fluorescent nanoparticles with combining PET and AIEE effects” for publication in Molecules.

We deeply appreciate you and referee’s kindness and patiently recommend us about the manuscript. The editor and reviewers’ comments are addressed in full as possibly as we can. First, we added several experimental results (Figure S2 Figure S6 and Figure S7) and redrawn Figure 2 and discussed some issues follow the suggestions from reviewers. Second, we corrected the mistakes, fulfilled some references for this time revision. All defences and revisions are marked with blue colour in manuscript and illustrated as follows.

I hope that you find this manuscript acceptable for publication in the article of Molecules. Please inform me if you have any other query and request.

Best regards,

Cheng-Chung Chang

8864-2284-0734#24,

ccchang555@dragon.nchu.edu.tw
